# A literature review on the analysis of symptom-based clinical pathways: Time for a different approach?

**Nammunikankanange Janak Gunatilleke**[1]*, **Jacques Fleuriot**[2], **Atul Anand**[3]

**1** College of Science and Engineering University of Edinburgh, Edinburgh, United Kingdom, **2** Artificial Intelligence and its Applications Institute, School of Informatics, University of Edinburgh, Edinburgh, United Kingdom, **3** Centre for Cardiovascular Science, University of Edinburgh, Edinburgh, United Kingdom

\* N.J.Gunatilleke@sms.ed.ac.uk

**Data Availability Statement:** All data are in the manuscript and/or supporting information files.

**Funding:** The author(s) received no specific funding for this work.

## Abstract

Breathlessness is a common clinical presentation, accounting for a quarter of all emergency hospital attendances. As a complex undifferentiated symptom, it may be caused by dysfunction in multiple body systems. Electronic health records are rich with activity data to inform clinical pathways from undifferentiated breathlessness to specific disease diagnoses. These data may be amenable to process mining, a computational technique that uses event logs to identify common patterns of activity. We reviewed use of process mining and related techniques to understand clinical pathways for patients with breathlessness. We searched the literature from two perspectives: studies of clinical pathways for breathlessness as a symptom, and those focussed on pathways for respiratory and cardiovascular diseases that are commonly associated with breathlessness. The primary search included PubMed, IEEE Xplore and ACM Digital Library. We included studies if breathlessness or a relevant disease was present in combination with a process mining concept. We excluded non-English publications, and those focussed on biomarkers, investigations, prognosis, or disease progression rather than symptoms. Eligible articles were screened before full-text review. Of 1,400 identified studies, 1,332 studies were excluded through screening and removal of duplicates. Following full-text review of 68 studies, 13 were included in qualitative synthesis, of which two (15%) were symptom and 11 (85%) disease focused. While studies reported highly varied methodologies, only one included true process mining, using multiple techniques to explore Emergency Department clinical pathways. Most included studies trained and internally validated within single-centre datasets, limiting evidence for wider generalisability. Our review has highlighted a lack of clinical pathway analyses for breathlessness as a symptom, compared to disease-focussed approaches. Process mining has potential application in this area, but has been under-utilised in part due to data interoperability challenges. There is an unmet research need for larger, prospective multicentre studies of patient pathways following presentation with undifferentiated breathlessness.

**Competing interests:** The authors have declared that no competing interests exist.

## Author summary

Breathlessness is a common symptom for patients attending hospital. This can be caused by many conditions and getting a diagnosis may be complex. The journey from attending hospital to a diagnosis is called a 'clinical pathway'. Process mining is a way of understanding the order and timing of events within a clinical pathway. We have reviewed the evidence for using process mining to better understand the clinical pathways for patients with breathlessness. We found 13 relevant studies and of those, only two focussed on the symptom and the rest on related diseases such as Covid-19. Most studies were also of small-scale, and the results were generally not replicated in more than one hospital. Only one study used true process mining to look at symptoms in patients attending a single Emergency Department. We believe that considering symptoms and using process mining could help clinicians understand how different patients interact with healthcare. This may improve the effectiveness of clinical pathways in the future, such as by reducing delays in diagnosis. Process mining may also help explain differences in response to treatments between individuals and help guide clinicians on the best time to organise tests such as specialist scans.

## Introduction

Breathlessness is a common presentation symptom to health services; it is undifferentiated and may be related to dysfunction in a wide range of body systems. This includes acute cardiac or respiratory conditions, infection, or metabolic disturbances. These are common presentations to acute hospitals. According to the Respiratory Care Action plan, more than one third of acute medical admissions to Scottish hospitals are due to respiratory diseases [1]. An analysis of National Health Service (NHS) Digital data of English hospital admissions revealed that 26–27% of all emergency episodes between 2011/12 and 2019/20, had a primary diagnosis with a high likelihood of breathlessness [2]. These numbers reduced to 21% in 2020/21, likely because of COVID-19 interventions on air pollutants [3,4].

The pandemic has accentuated the critical importance of efficient clinical pathways. A clinical pathway is a standardised set of processes that can be used to improve the efficiency of healthcare [5]. These are commonly defined around disease states such as asthma and heart failure, but there is much more complexity in defining clinical pathways for patients presenting with undifferentiated symptoms. For example, breathlessness is a core feature of severe COVID-19, but cardiovascular, gastrointestinal, and neurological body systems may also be affected by the virus [6]. This highlights the challenge facing clinicians in care settings such as an Emergency Department (ED). Patients may transition between primary and secondary care with multiple clinical interactions for persistent symptoms before confirmation of a diagnosis. Therefore, clinical pathway analysis using diagnostic codes may fail to recognise the barriers of a prolonged pre-diagnostic phase that may influence later outcomes.

Electronic Health Records (EHRs) have become mainstay of clinical care in the last two decades. A recent English study reported that 77% of acute healthcare trusts were using an EHR system [7]. While this opens up new possibilities to understand care pathways, symptom-based coding remains rare, necessitating natural language processing to extract data from free-text entries [8]. A further challenge is the lack of interoperability of EHR systems, which may prevent aggregation of data across providers [9]. The majority of studies in this area are based at a single centre, which limits the generalisability of findings [10].

Outside of healthcare, process mining is frequently used to understand pathways for navigating through complex systems e.g. in business settings to include analysing the invoicing of subcontractors and supplier, and identifying fraud at a leading European financial institution [11,12]. It describes the discovery, monitoring, and improvement of real (and not assumed) processes by extracting knowledge from event logs, such as those readily available in electronic health record systems [13]. Process mining can help extract information (for example, investigations and treatments a patient received and when they received it) to visualise the various workflows, or pathways, related to a patient diagnosis or treatment in a hospital. There are several algorithms that can be used in process mining such as the alpha miner, heuristics miner, fuzzy miner, and inductive miner [13,14]. There is a balance between the following four aspects of quality when considering the validity of the output model from the process mining algorithms: fitness (how accurately can it reproduce the behaviours in the event log?), precision (how many behaviours does it allow that is not in the event log?), generalisation (can it reproduce behaviours that are relevant but not specifically included in the log?), and simplicity (how easy is the model to read?) [13,15].

In this study we sought to answer the question–how have process mining or related techniques been used to analyse clinical pathways for patients presenting with breathlessness? We selected breathlessness due to its frequency and importance as a symptom in patients seeking medical attention. An understanding of the current state will inform opportunities to use process mining and a symptom focused approach to analyse clinical pathways, with the ultimate aim of improving patient outcomes and healthcare service delivery.

In this literature review, we synthesise the evidence for the application of process mining to the analysis of clinical pathways for patients with breathlessness. We approach this in two ways. First, we review studies taking a symptom-focussed approach for patients presenting with breathlessness. Second, we review studies taking a disease-focussed approach including cardiorespiratory conditions that commonly present with breathlessness.

## Materials and methods

### Search strategy

We conducted searches for relevant material in PubMed, IEEE Xplore, and the ACM Digital Library. These were chosen to combine the leading medical research database with collections focussed on relevant technical, process mining and computer science topics. Complementary searches were completed using a specific database of process mining studies–processsmining. org–and using forward citation searches of recent literature reviews to identify any further relevant studies. A wide search strategy was utilised to maximise the identification of potential studies. The searches were conducted in October and November 2021.

Analysis of recent literature reviews conducted on process mining studies in healthcare [10,16,17] informed the development of the search terms to cover process mining and variants. These included 'process mining', 'workflow mining', 'pathway analysis' and 'data mining'. In the first searches these terms were combined with synonyms for breathlessness (*S1 Appendix*). The second search joined process mining terms with respiratory and cardiovascular disease terms (*S2 Appendix*).

Similar searches were performed on processsmining.org and using forward citation searches for three recent process mining reviews in healthcare. [10,16,17]

### Selection criteria

Studies were included if breathlessness, a similar state (e.g. respiratory distress) or relevant disease was present in combination with a process mining concept. Articles were excluded if

symptoms or relevant diseases were not studied, such as a focus on biomarkers, if not in English, if not primary research of a dataset (e.g. reviews, opinion pieces), or in the absence of a clinical pathways analysis. No restrictions were placed on publication time periods. Initial screening was conducted using title and abstract review, with eligible studies undergoing full text review.

## Results

A total of 1,385 studies were found with 1,324 studies through PubMed, 14 through IEEE Xplore and 47 through ACM Digital Library searching. A total of 523 of studies were found through complementary searching (including forward citations) and initial screening resulted in the 15 studies included for further review. After exclusion of duplicates and abstract screening, 68 full text articles were reviewed, of which 13 were included in this review (*Fig 1*).

The majority of screened abstracts were disease focussed (1318/1376, 96%). The main reasons for excluding studies during full text review were a lack of symptom focus (e.g. reports of alternative symptoms, investigations, or risk factors), considering patients already diagnosed with a condition, systematic or other reviews of studies, and focus on diseases other than those defined in the search strategy. One study was excluded for describing a theoretical approach to clinical pathways, without actual evidence of implementation [18]. The 13 studies included in the final synthesis of evidence are summarised in *Table 1*.

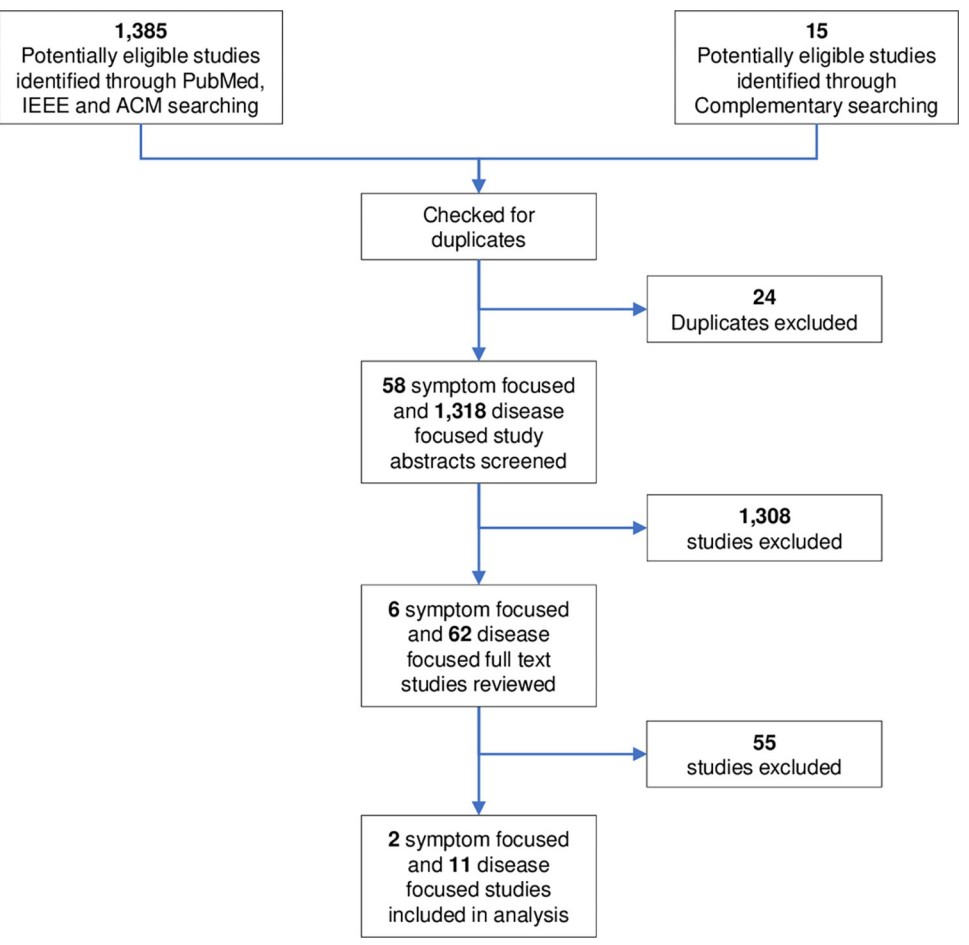

**Fig 1. Summary of study selection.**

**Table 1. Summary of included studies.**

| Author | Context | Techniques and methods used | Summary of key findings |
|---|---|---|---|
| **Symptom focus** | | | |
| Duma *et al* (2020) [19] | Retrospective analysis of over 88,000 patients attending one Italian ED. | Decision trees to cluster patients and Hybrid Activity Tree notation to model the processes | • Aim was to develop a process model to identify possible patient pathways in the ED and to identify next activities.<br>• The resulting model proved to be better suited for the complexity of ED patient paths than traditional process mining approaches such as Heuristic miner.<br>• Further research applying the methodology to other Eds would be needed to fully validate the approach. |
| Grigull *et al* (2012) [20] | Retrospective and prospective (testing) analysis at an ED in one hospital in Hannover. | Algorithms including ANNs, SVMs, and fuzzy logic (a voting algorithm was used to combine the outputs of the three algorithms) | • Aim was to demonstrate the capacity of a combination of different data mining (DM) methods to support diagnosis in undifferentiated paediatric emergency patients.<br>• The combination of four DM operations arrived at a correct diagnosis (retrospectively) in 98% of the cases, with the highest diagnostic accuracy for appendicitis and idiopathic thrombocytopenic purpura or erythroblastopenia.<br>• During a prospective testing phase, 81% of the patients were correctly diagnosed by the system. |
| **Disease focus** | | | |
| Alizadeh *et al* (2015). [21] | Retrospective analysis of a database for data from one department—lung clinic—in Tehran. | Neural networks | • Aim was to use data mining techniques to aid diagnosis of asthma.<br>• 13 of the 22 recorded factors were relevant for the diagnosis, and the system was able to predict diagnosis in 100% of cases.<br>• Optimising input data, including reducing data dimensionality, can improve accuracy.<br>• The study only included 254 patients with complete records for both testing and validation limiting generalisability. |
| Alizadehsani *et al* (2012) [22] | Retrospective analysis within one hospital. | C4.5, Naïve Bayes, and k-nearest neighbours (kNN) algorithms | • Aim was to predict the likelihood of coronary artery stenosis from risk factors, physical examination, and clinic data.<br>• Age and typical chest pain were the key factors having the highest influence on stenosis, with the C4.5 decision tree having the optimum performance.<br>• The small sample size of 303 patients and lack of external validation limits the impact of findings. |
| Alizadehsani *et al* (2013) [23] | Retrospective analysis within one hospital. | Classification algorithms including SMO, Naïve Bayes, Bagging algorithm and Neural network algorithm | • Aim was to assess Coronary Artery Disease prediction capability using classification algorithms.<br>• The highest accuracy was achieved by the Sequential minimal optimization (SMO) algorithm along with the feature selection and feature creation methods.<br>• The accuracy of 94.8% with SMO was the highest reported in published research<br>• The small sample size of 303 patients and lack of external validation limits the impact of findings. |
| Haug *et al* (2013) [24] | Retrospective analysis of Clinical Data Warehouse for ED data in two hospitals. | Bayesian network classifier (and NLP to extract insights from free text reports) | • Aim was to take a medical ontology-based approach to automate the creation of diagnostic decision-support applications.<br>• The ontology-based solution performed well against a manually created tool to support pneumonia diagnosis (0,92 area under the receiver operating characteristic curve with automated tool compared to 0.944 with manually created tool). |

*(Continued)*

**Table 1.** (Continued)

| Author | Context | Techniques and methods used | Summary of key findings |
|---|---|---|---|
| Langer *et al* (2020) [25] | Retrospective analysis of Patient Data Management Systems in one hospital in Milan. | A number of algorithms including ANNs (D_FF variants and MLP) Random Forest, Naïve Bayes, Rotation Forest, Logistic, Logit Boost, J48, SMO and kNN | • Aim was to assess the accuracy of artificial intelligence in predicting the results of RT-PCR (Reverse transcription polymerase chain reaction) for SARS-COV-2, using basic information commonly available at ED presentation.<br>• The TWIST (Training with Input Selection and Testing) approach helped D_FF_Conic reach an accuracy of 91.4% with 94.1% sensitivity and 88.7% specificity.<br>• The study was limited by small sample size of just 199 patients without external validation, and therefore larger multicentre studies would be needed to confirm potential of using basic clinical data to predict Covid-19 without needed RT-PCR |
| Leidy *et al* (2016) [26] | Retrospective analysis of 3 chronic lung disease databases in the United States. | Random Forest Analysis | • Aim was to develop a practical and effective primary care strategy for identifying undiagnosed patients with clinically significant Chronic Obstructive Pulmonary Disease (COPD).<br>• Between four and eight variables were identified to differentiate cases and controls with error rates of between 9% and 27%.<br>• Results suggested that helping patients recognise breathing-related activity impairments could help identify more patients who would benefit from treatment. |
| Marin-Gomez *et al* (2021) [27] | Retrospective analysis of EHR data from 311 Primary Care Teams (PCT) run by the Catalan Institute of Health. | Multinomial logit model and QUEST (Quick, Unbiased and Efficient Statistical Tree) classification type of tree | • Aim was to evaluate the application of a decision tree to the diagnosis of Covid-19 based on clinical and sociodemographic information.<br>• The decision tree had only moderate sensitivity and specificity of 64.3% and 62.5% respectively.<br>• 'Contact with a suspected/confirmed case' was the best predictor suggesting value of analysing contact information. |
| Perotte *et al* (2021) [28] | Retrospective analysis in EHR (EPIC) in one ED in New York | Hierarchical Clustering Analysis. | • Aim was to evaluate chief complaints of patients presenting to ED with suspected or confirmed Covid-19.<br>• Seven complaint clusters (complaints that occur together) and age-based differences were identified.<br>• Shortness of breath was the most commonly identified symptom in the Covid-19 positive group (second most common across entire study population) |
| Rother *et al* (2015) [29] | Retrospective analysis via answering a questionnaire in one hospital in Hannover. | Eight classifiers including Support vector machines (SVMs), Artificial neural networks (ANNs), fuzzy-rule based, random forest, logistic regression, linear discriminant analysis, Naive Bayes and nearest neighbour, and a fusion of all eight | • Aim was to develop and test a parental observation based questionnaire and data mining-supported tool providing diagnostic support for three rare childhood pulmonary diseases.<br>• The fusion algorithm detected the correct diagnoses with an overall sensitivity of 98.8%.<br>• The potential value of the tool in raising awareness (and potentially reducing diagnostic delay) for rare pulmonary diseases such as PCD (Primary ciliary dyskinesia) or CF (Cystic Fibrosis) was highlighted.<br>• The study was limited by small sample size and internal cross-validation. |

*(Continued)*

**Table 1.** (Continued)

| Author | Context | Techniques and methods used | Summary of key findings |
|---|---|---|---|
| Shanbehzadeh *et al* (2021) [30] | Retrospective analysis of patients presenting for Covid-19 treatment in one hospital in Iran. | Decision tree algorithms including J-48, Decision stump, Hoeffding Tree, LMT, Random forest, Random tree and REP-tree | • Aim was to compare seven decision tree algorithms to select the best clinical diagnostic model for Covid-19.<br>• J-48 (an open source version of a C4.5 decision tree algorithm) performed the best in terms of accuracy, F Score, Receiver Operator Characteristic (ROC) and Precision Recall Curve (PRC) for Covid-19.<br>• J-48 can potentially be used to increase the efficiency of the diagnosis of Covid-19.<br>• The small sample size of 400 patients and lack of external validation limits impact of findings. |
| Vijayakrishnan *et al* (2014). [31] | Retrospective analysis of EHR data in 41 community practice clinics. | Natural Language Processing (NLP) | • Aim was to identify the documentation of the signs and symptoms of heart failure in the years preceding its diagnosis.<br>• Signs and symptoms including breathlessness were identified years before heart failure diagnosis, suggesting potential for using EHR data in predictive models<br>However, these were also present in the majority of controls highlighting a lack of specificity of this approach without additional clinical data. |

## Discussion

We have reviewed the use of process mining in clinical pathway analysis of patients with breathlessness, identifying some key barriers. First, symptom-based approaches to analysing clinical pathways for this important and common presentation are rare compared to those focussed on specific diseases. Of 13 studies included in this review, only two considered patients with undifferentiated symptoms. Second, a variety of data mining and machine learning approaches have been explored, but process mining has rarely been used in the analysis of these patient pathways, with just one study employing this technique. Third, clinical pathway analyses are generally trained and internally validated within single-centre datasets, limiting evidence for wider generalisability. Taken together, these findings suggest an opportunity to further explore process mining as a tool to understand clinical pathways for undifferentiated patients with breathlessness.

Our findings are in keeping with a prior review in the broader area of process mining in healthcare, which highlighted a lack of focus on respiratory diseases or symptoms. Across 152 studies in this area, only 3% included respiratory conditions, and just 2% were symptom orientated [10]. Similar findings were present in an older literature review of 67 studies [16]. It is unsurprising that four recent studies identified in our review focussed on COVID-19. The pandemic has brought the importance of symptom-based triage into sharp focus, as public health measures attempt to identify and isolate possible infective cases based on pre-test probability [32]. However, for studies focussing purely on a disease such as COVID-19, a cluster-based analysis is likely to be required to capture all potential symptoms beyond breathlessness, which may only occur in severe disease. Indeed, the study by Perotte *et al* [28] included in our review identified seven distinct COVID-19 symptom clusters of which only one included shortness of breath.

We set out in this review to identify examples of process mining methodology applied to breathlessness clinical pathways, accepting that our search included other data mining approaches. The three recognised components of process mining are process discovery (e.g. identifying the processes within a hospital department), conformance checking (e.g.

comparing the management of a disease against published best practice guidance) and enhancement (e.g. identifying opportunities to improve a pathway or process within a hospital or department in terms of outcomes or efficiency). Only one study by Duma *et al.* truly applied process mining techniques, using a dataset of nearly 90,000 Emergency Department attendances classified by primary presenting symptom [19]. The authors ultimately applied a variety of machine learning algorithms including decision trees, random forests, k-nearest neighbours (kNN), artificial neural networks (ANN), support vector machines (SVM) and naïve Bayes [33]. This followed an exploration of the dataset using standard process discovery methods (Heuristic Miner [34] and Inductive Miner-infrequent [14]), which lacked either model fitness or precision to explain the activity observed in event logs. Instead, the authors used 'ad hoc' process discovery, attempting to adequately balance fitness, precision, generalisation, and simplicity.

In studies that did not meet inclusion criteria for this review, other authors have explored similar challenges in applying process mining to health datasets. Najar *et al.* [35] describe a two-step approach to complexity and variability, by first clustering the data into similar groups before applying process mining techniques. Process mining has been shown to be an input to and support validation of discrete event simulation [36], which attempts to decompose a complex pathway into logically separated processes that occur in a sequence over time. Developing these models could also provide additional support in developing health AI solutions. For example, it could provide more structured and refined outputs for AI model training, a better way to validate AI model performance and improve the ability to build more inclusive AI models that account for variance and outliers.

A multimodal approach to data mining was also employed by the other symptom-based study identified in this review [20]. It is plausible that undifferentiated symptom-based approaches as we have proposed, introduce such high heterogeneity in data that more than one technique is required to accurately explain the complexity of clinical pathways. This may in part explain the predominance of disease focussed studies, as the commonality of a single condition across a study population dramatically reduces variation in data. This can be taken even further by ontology-based approaches that predefine categories of data including their interdependencies, to provide a common language for the structure of a dataset and concepts within it. The study by Haug *et al* [24]. successfully used such a system to simplify the analysis of a large dataset for the identification of pneumonia in the Emergency Department. While this technique appears attractive, it requires significant data conformity that does not clearly translate into undifferentiated patient pathways like breathlessness.

Population health and outcome based health systems have been explored and, in England, Integrated Care Systems (ICS) are due to be fully operational from April 2022. Additionally, Covid-19 has highlighted the challenges of taking a purely activity-based approach to healthcare. It is expected that there will be an increased focus on efficient and outcome-focussed care pathways across care and community settings that maximise the health and wellbeing of citizens. This will also provide incentives to develop interoperability across healthcare information systems, adopt technology standards such as Fast Healthcare Interoperability Resources (FHIR) that can enable process mining [37] and to standardise and share data across organisations. The majority of studies included in this review used a single healthcare setting, with only four multicentre studies identified. There is an urgent need for improved interoperability of healthcare systems to facilitate large scale research. Recent national initiatives such as streamlined funding arrangements and benchmarking tools [38], funding allocations such as the Digital Aspirants programme [39] and increased government financial support for NHS digital and technology deployment are expected to increase prevalence and functionality of EHRs.

Given these data challenges, what are the potential benefits of a process mining approach for understanding clinical pathways based on patient symptoms? First, it could reduce the delays for patients with recurrent symptoms who do not meet the full diagnostic criteria for specific diseases. For example, the diagnosis of pulmonary embolus is challenging due to varied symptomology at presentation, sometimes but not always including breathlessness. This can cause delays in diagnosis as investigations for other more common respiratory and cardiac conditions may be prioritised. Defining and understanding the clinical pathways associated with delayed diagnosis for pulmonary embolus, could inform clinician education in this area and help identify patients at risk of a missed or delayed diagnosis [40,41].

Second, process mining could help improve understanding of patient outcomes for different pathways and the associated costs. Arias *et al.* [42] used process mining to develop a patient journey map for the diagnosis of pneumonia. Through this, the authors identified the most common pathway (which covered 28% of the cases) and the associated six key patient touchpoints. Furthermore, they were able to identify relationships between age, gender, and length of stay. This information could be used to optimise treatment interactions or identify system weaknesses in particular pathways. Dahlin *et al.* [43] used process mining to explore breast cancer patient pathways across four Swedish hospital groups. When considering the cost per patient during the 14 months following diagnosis, they were able to highlight significant differences across four patient groups, including one example pathway that appeared more cost effective than the most frequently observed pathway.

Third, process mining could help identify what level of investigation and intervention is most appropriate at time of presentation to deliver safe but efficient care. Earlier broader investigations for a cause of undifferentiated breathlessness, such as high-resolution CT scanning, may be appropriate for some patients, but this is a finite resource and includes radiation exposure that would be inappropriate if applied uniformly. Similarly, understanding which patients can be safely managed using a 'watch and wait' approach to their symptoms could improve efficiencies of care.

## Limitations and future research

We have deliberately performed a focussed literature review in this area. Wider use of multiple search databases could have yielded more relevant results, as could an increase in the number of diseases specifically included within the search strategy. However, the limited number of published studies in this area is in keeping with prior wider reviews of process mining in healthcare.

We did not perform a formal systematic literature review and therefore did not undertake a formal quality assessment using a structured tool. However, we have highlighted the quality of included studies where relevant.

In general, the studies identified were small and largely based on retrospective single-centre datasets; internal cross-validation was predominantly used over external validation, which limits the generalisability of findings. Given the heterogeneity of patient data in this area, it is unlikely that datasets of a few hundred patients from a single centre could derive meaningful clinical decision support tools. To fully evaluate the potential highlighted in using process mining and related techniques in analysing clinical pathway, we would suggest that future research in this area focus on larger cohorts and multicentre studies, and that researchers externally validate their findings.

## Conclusions

Our review has highlighted a lack of clinical pathway analyses for breathlessness as a symptom, compared to more traditional disease-focussed approaches. Process mining featured in only

one study, but has clear potential application to this area. Progress has been hampered by data interoperability challenges, but the potential for improved care through analyses of clinical pathways is increasingly recognised at a policy and funding level, in part due to learning from the COVID-19 pandemic. There is an unmet research need for larger, prospective multicentre studies of patient pathways following presentation with undifferentiated breathlessness.

## Supporting information

**S1 Appendix. Search terms used for PubMed search.**
(TIF)

**S2 Appendix. Search terms used for IEEE Xplore and ACM Digital Library searches.** To make the search work on IEEE Xplore and ACM Digital Library, the [Title/Abstract] restriction was removed. To counter a significant increase in search results, the search term 'Data mining' was also removed.
(TIF)

## Author Contributions

**Conceptualization:** Nammunikankanange Janak Gunatilleke.

**Formal analysis:** Nammunikankanange Janak Gunatilleke.

**Investigation:** Nammunikankanange Janak Gunatilleke.

**Methodology:** Nammunikankanange Janak Gunatilleke.

**Supervision:** Jacques Fleuriot, Atul Anand.

**Visualization:** Nammunikankanange Janak Gunatilleke.

**Writing – original draft:** Nammunikankanange Janak Gunatilleke.

**Writing – review & editing:** Jacques Fleuriot, Atul Anand.

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
